

# Merging of a mesoscale eddy into the Lofoten Vortex in the Norwegian Sea captured by an ocean glider and SWOT observations

Gillian M. Damerell[1], Anthony Bosse[2], and Ilker Fer[1]

[1]Geophysical Institute, University of Bergen, and Bjerknes Centre for Climate Research, Bergen, Norway
[2]Aix Marseille Univ, Université de Toulon, CNRS, IRD, MIO, Marseille, France

**Correspondence:** Gillian M. Damerell (gillian.damerell@uib.no)

**Abstract.** The Lofoten Vortex (LV) is an intense, apparently permanent anticyclone in the Lofoten Basin of the Norwegian Sea. It is characterised by a 1200 m thick core of Atlantic Water, with a radius of 15-20 km, in nearly solid-body rotation reaching speeds up to 0.8 m s$^{-1}$. Potential vorticity in the core is nearly two orders of magnitude lower than the surroundings, creating a barrier to lateral mixing. It has previously been postulated that anticyclonic eddies in the Lofoten Basin, shed from the eastern branch of the Norwegian Atlantic Current along the Lofoten Escarpment, merge into the LV, contributing to maintaining its large heat and salt content and energetics, but such merging events have proven difficult to observe directly due to their transient and unpredictable nature. In April 2023, an eddy merger event was successfully observed using a combination of in-situ data from an autonomous ocean glider and absolute dynamic topography (and derived velocities) from the fast sampling calibration phase of the Surface Water Ocean Topography (SWOT) satellite altimeter. During the observed merging process an incoming eddy gradually approaches the LV, then elongates as the two begin to corotate and then merge, with a corresponding spin up of vorticity and eddy kinetic energy and possible ejection of low potential vorticity water from the merged LV core. The incoming eddy had a smaller radius and higher Rossby number than the LV. It has a similar density range as the LV and therefore a double-core vertical structure did not form after the merger. During the observed period, merging eddies were the dominant process affecting the evolution of the LV, clearly outweighing vertical 1D processes due to atmospheric forcing and lateral mixing between the LV core and the outer rim. Through influx of buoyant waters, spin-up of eddy kinetic energy and increasingly anticyclonic vorticity, eddy mergers contribute to the longevity of the LV.

## 1 Introduction

The Lofoten Basin is a topographic depression with a maximum depth of 3250 m, located in the Norwegian Sea. The basin lies between the Norwegian continental slope in the east, the Vøring Plateau and the Helgeland Ridge in the south and south-west, and the Mohn Ridge in the north-west (Fig. 1). It is the largest oceanic reservoir of heat (Rossby et al., 2009a; Bosse et al., 2018) in the Nordic Seas (a common collective name for the Greenland, Iceland and Norwegian Seas), and plays an important role in regional and global climate dynamics (Segtnan et al., 2011; Asbjørnsen et al., 2019; Broomé et al., 2020; Brakstad et al., 2023). The Norwegian Atlantic Current, a branch of the North Atlantic Current, brings warm Atlantic Water poleward in two branches through the Iceland-Faroe Ridge and the Faroe Shetland Channel (Poulain et al., 1996; Orvik and



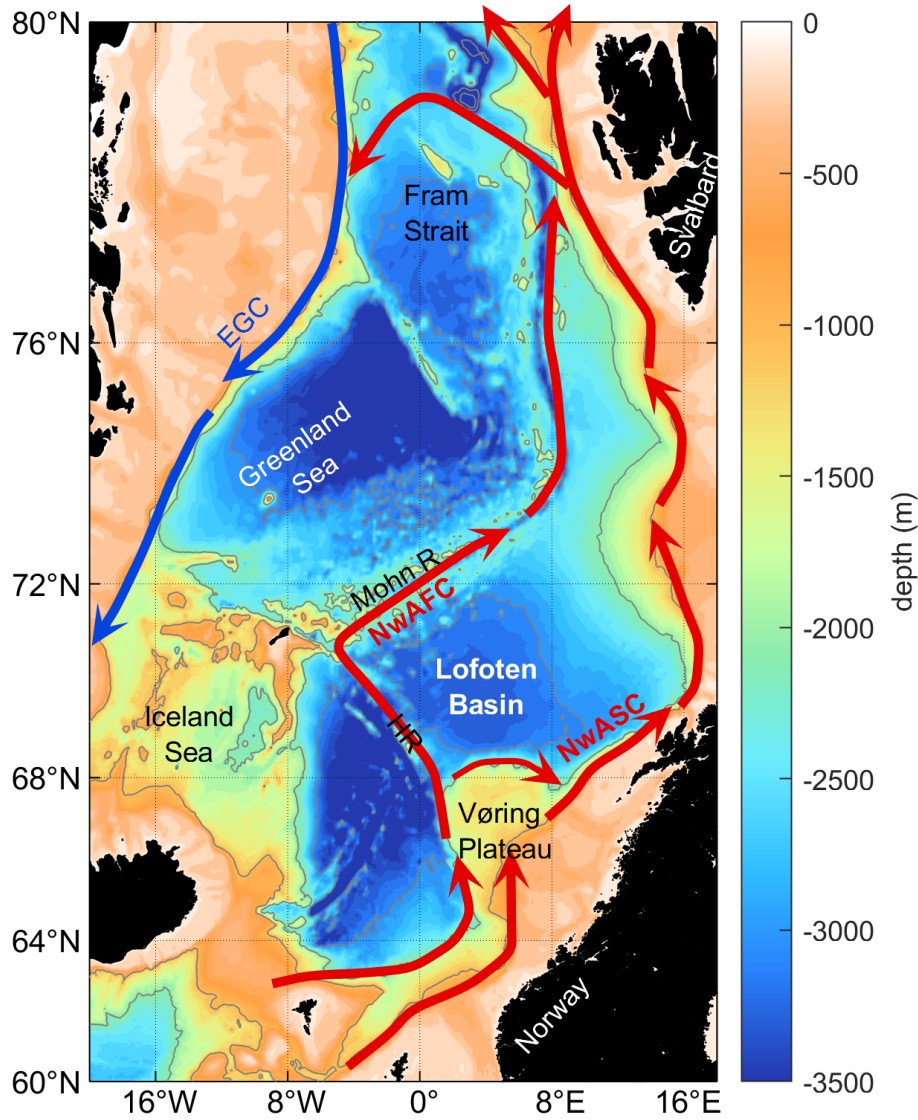

**Figure 1.** The Nordic Seas. Bathymetry is shown in colour, with grey contour lines every 1000 m. The coloured arrows represent the main circulation patterns, with the warm Norwegian Atlantic Current in red and cold East Greenland Current (EGC) in blue. HR = Helgeland Ridge, Mohn R = Mohn Ridge, NwAFC = Norwegian Atlantic Front Current, NwASC = Norwegian Atlantic Slope Current.

Niiler, 2002; Rossby et al., 2009b). The latter forms the Norwegian Atlantic Slope Current (NwASC), which flows northward as a barotropic shelf-edge current along the continental shelf of Norway (Orvik and Niiler, 2002). The outer baroclinic branch forms the Norwegian Atlantic Front Current (NwAFC), which flows along the western flank of the Vøring Plateau and then along the Helgeland and Mohn ridges (Orvik and Niiler, 2002; Bosse and Fer, 2019). The Lofoten Basin, located between these two current systems, contains relatively warm waters for this latitude.



Warm Atlantic Water enters the Lofoten Basin in two ways: from the south as the result of the separation of the two branches of the Norwegian Atlantic Current (Rossby et al., 2009b; Dugstad et al., 2019), and from the east via coherent anticyclonic mesoscale eddies shed from the NwASC near the Lofoten Escarpment (Köhl, 2007; Isachsen, 2015; Richards and Straneo, 2015; Volkov et al., 2015; Fer et al., 2020). Fer et al. (2020) analysed 14-month long mooring records from the Lofoten Escarpment and reported highly energetic current variability reaching $\pm 0.4$ m s$^{-1}$ in the 35 hour - 14-day band, suggesting

substantial eddying activity. They supplemented limited mooring-based energy conversion estimates with results from a high-resolution, eddy-resolving regional model. The model used was the Regional Ocean Modelling System (ROMS), a hydrostatic model with terrain-following coordinates with a horizontal resolution of 800 m and 60 vertical layers. The NwASC has a large baroclinic component along the Lofoten Escarpment, and Fer et al. (2020) demonstrated that the baroclinic energy conversion rates are typically positive and quite large along the slope of the Lofoten Escarpment, indicating that the baroclinic instability of

the slope current extracts energy from the mean flow and converts it to eddy kinetic energy. After formation, eddies propagate westward and transport heat towards the deepest part of the Lofoten Basin (e.g., Rossby et al., 2009b; Andersson et al., 2011). Raj et al. (2016) used satellite data from 1995 to 2013, together with data from surface drifters and Argo floats, to characterise the eddy field of the Lofoten Basin, and inferred a generally cyclonic drift of eddies in the western Lofoten Basin with mean speeds of 5-6 km d$^{-1}$, consistent with previous analysis of the mid-depth circulation from Argo floats trajectories by Voet et al.

(2010). The eddy lifespan varied from days to months for both cyclonic and anticyclonic eddies, but a greater portion of the anticyclonic eddies were long-lived, and these long-lived anticyclonic eddies were found predominately in the western Lofoten Basin around the Lofoten Vortex (LV).

    The LV, also known as the Lofoten Basin Vortex or Lofoten Basin Eddy, is an apparently permanent oceanic anticyclonic eddy located in the deepest part of the Lofoten Basin (Søiland and Rossby, 2013; Yu et al., 2017). First observed in the 1970s

(Ivanov and Korablev, 1995a), the LV consists of a core of warm, saline Atlantic Water which can reach as deep as 1200 m (Bosse et al., 2019). The core is surrounded by a region of intense azimuthal velocities. These azimuthal velocities can reach 0.8 m s$^{-1}$ at 600-800 m depth and relative vorticities are typically around $-0.5f$ but sometimes reach as low as $-0.9f$, close to the theoretical limit for anticyclones (Bosse et al., 2019; Fer et al., 2018). The radius of the LV core is usually defined as the radius at the maximum azimuthal velocity, $R_v$, and has typically been found between 15 and 20 km by current measurements

from ADCPs and glider cyclogeostrophic reconstruction, as well as Lagrangian RAFOS floats drifting at 500-800 m depth (Søiland and Rossby, 2013; Yu et al., 2017; Fer et al., 2018; Bosse et al., 2019). Raj et al. (2015) used a persistent sea level anomaly of at least 2 cm to detect the LV in 16 years of satellite altimetry data from 1995 to 2010, and found the radius of the LV to have a mean value of 37 km, but gridded satellite altimetry products are known to overestimate the LV radius as defined by the maximum azimuthal velocity (Yu et al., 2017). They also found the LV centre to be located between $69.5 - 70°$N,

$2.5 - 4.1°$E (1st and 3rd quartile). This is consistent with the LV locations found by Søiland and Rossby (2013); Yu et al. (2017) and Bosse et al. (2019). The persistence and location of the LV likely influence the water mass transformations in the region: within the LV Atlantic Water has been found at depths several hundred metres deeper than in surrounding parts of the Lofoten Basin (Raj et al., 2015; Bosse et al., 2018).



Several authors have reported a multiple-core vertical structure in the LV, i.e., weakly stratified cores separated by layers of
higher stratification, although single cores have occasionally been observed (e.g., Søiland and Rossby, 2013; Raj et al., 2015;
Søiland et al., 2016; Yu et al., 2017; Fer et al., 2018; Bosse et al., 2019). Yu et al. (2017) and Bosse et al. (2019) discuss the
varying vertical structure in terms of seasonal changes, using ocean glider observations of the LV taken between 2013 and
2015, and 2016-2017 respectively. During winter, strong cooling at the surface results in convective mixing reaching several
hundred metres deep (though not as deep as 1000 m in the years they observed), then beginning in approximately April/May,
surface warming creates a cap of restratified waters above the core which can extend down to approximately 200 m over the
summer. When wintertime convection resumes, the upper water column is again homogenised as the mixed layer grows. As
time progresses, the temperature and salinity properties of each wintertime core layer change gradually, the layers densify and
are found deeper in the water column. The layer formed each winter is found above that of the previous winter, resembling
vertical "stacking" (see, for example, Yu et al.'s Fig. 11). However, while the observations of Yu et al. (2017) suggest wintertime
convection penetrating to a few hundred metres at most, Søiland and Rossby (2013) and Søiland et al. (2016) found highly
homogenised water to ∼1000 m in July 2010 and September 2012, suggesting that convection penetrated to at least that depth
during the preceding winter.

Based on ocean microstructure observations from a summer cruise, Fer et al. (2018) estimated the time scale to drain the
volume-integrated total energy of the LV as $\mathcal{O}(10)$ years. The turbulent dissipation rate was driven by strong shear below the
swirl velocity maximum at the vortex rim, and near-inertial waves trapped by the negative vorticity in the vortex core. Bosse
et al. (2019), however, considered additional processes of surface and bottom drag and found this time scale to be only 3
years. The long-lived LV must therefore be maintained by important processes which supply potential energy that overcomes
the energy dissipation beyond a few-year time scale. Two mechanisms have been proposed to play a role in maintaining the
structure and characteristics of the LV: wintertime deep convection (Ivanov and Korablev, 1995a, b), and merging of mesoscale
eddies into the LV (Köhl, 2007). Convection events are hypothesised to deepen the isopycnals below the vortex core, increase
the radial density gradient, and intensify the azimuthal velocity. In an idealised numerical case study, de Marez et al. (2021)
conclude that the LV can survive thanks to a balance between merger, convection, and bottom drag. Trodahl et al. (2020)
used the same eddy-resolving model as Fer et al. (2020) (ROMS, horizontal resolution 800 m, discussed above), to study the
mesoscale eddy merging process, and identified 3–4 merger events each year with no clear seasonal bias. This is in contrast
to two previous modelling studies, both using the Massachusetts Institute of Technology general circulation model with a
horizontal resolution of 4 km: that of Köhl (2007), who also found three to four merging events per year but with slightly
more mergers occurring during the period February–May and none in November–December, and the study by Belonenko et al.
(2017), who identified one to two mergers per year with a distribution skewed toward wintertime. Observations by Bosse et al.
(2019) identified a possible eddy merger in winter and described the seasonal weakening of Potential Vorticity (PV) gradients
between mesoscale eddies and the LV core in the upper layer as more favourable to eddy mergers.

While the supporting role of convection cannot be ruled out, Trodahl et al. (2020) conclude that the LV is mainly maintained
through repeated merging events. Incoming eddies approach the LV, begin to corotate with it, become elongated and wrap
around the LV, and are eventually absorbed by it. The LV is typically denser at its core than incoming eddies because the LV is



subject to prolonged cooling periods whereas the incoming eddies are shorter-lived and have not had time to lose as much heat
since they were spun out from the NwASC. Thus the merging process modelled by Trodahl et al. (2020) frequently results in a double-core vertical structure at the end of the merging process, which can persist for weeks to months, because the lighter incoming eddy is stacked above the original LV core (see Trodahl et al.'s Fig. 14). This accords well with the eddy merging process observed by Garreau et al. (2018) in the Mediterranean Sea, Belkin et al. (2020) in the Gulf Stream and Rykova and Oke (2022) in the Tasman Sea, who also found that the merging of two eddies of different density led to the stacking of the lighter eddy above the denser one in the resulting combined eddy. In the LV, the cores of the vertically aligned denser vortex and the lighter merging eddy are subsequently fused through vertical convection (Trodahl et al., 2020). However when the incoming eddy is of a similar density to the LV, the result is a single LV core. During the merging process, the combined vortex becomes more vertically stratified, intensifies by compression and PV conservation, and expels some low-PV fluid. Wintertime convection serves mainly to vertically homogenise and densify the LV impacting its potential energy, whereas merging events have the strongest impact on intensifying it by transfer of kinetic energy.

In December 2022, the Surface Water and Ocean Topography (SWOT) satellite was launched. It embarks a Ka-band radar interferometer (KaRIn) providing maps of Sea Surface Height (SSH) at a native horizontal resolution of 250 m and noiseless products at 2 km within two swathes each 60 km wide separated by a 20 km gap; in the middle of the gap, sea level is measured by a conventional nadir altimeter (Fu et al., 2024). The SWOT satellite's fast sampling calibration and validation phase (henceforth the Cal/Val phase) lasted from 29 March to 10 July 2023: during this time the satellite passed over the same locations on the planet each day. One of the crossover points was located above the mean position of the LV (Fig. 2), thus this area was observed twice per day, once on the ascending track and once on the descending track. SWOT offers a significant increase in resolution over nadir altimeters such as Jason, Sentinel and TOPEX/Poseidon. While such satellites have improved our knowledge of mesoscale eddies with horizontal scales $O(50\text{-}500)$ km, they can miss smaller mesoscale eddies and nearly all submesoscales (Klein et al., 2019). Zhang et al. (2024) have demonstrated that SWOT is easily able to resolve submesoscale eddies with radii as small as 16 km and sea level anomaly amplitudes as small as 2 cm, thus it should have no difficulty resolving the LV. This unprecedented resolution offered the opportunity to observe the LV, and nearby eddies, as never before, and thus coincident glider missions were planned to collect observations from the LV during the SWOT Cal/Val phase, participating into the global effort of in situ data collection during this crucial phase (d'Ovidio et al., 2019).

In this paper, we present in-situ observations of a mesoscale eddy merging into the Lofoten Vortex, which took place between 5 - 16 April 2023. Subsurface observations are supplemented by the measurements from SWOT. Section 2 presents the data and methods, section 3 the results and discussion, and section 4 the conclusions.

## 2    Data and Methods

### 2.1    In situ data

Hydrographic data were acquired using the Seaglider SG563 which was deployed from RV *Johan Hjort* on 19 January 2023, and recovered on 8 June 2023 by RV *G.O. Sars*, completing 620 dives in total. Seagliders are small, autonomous, remotely-



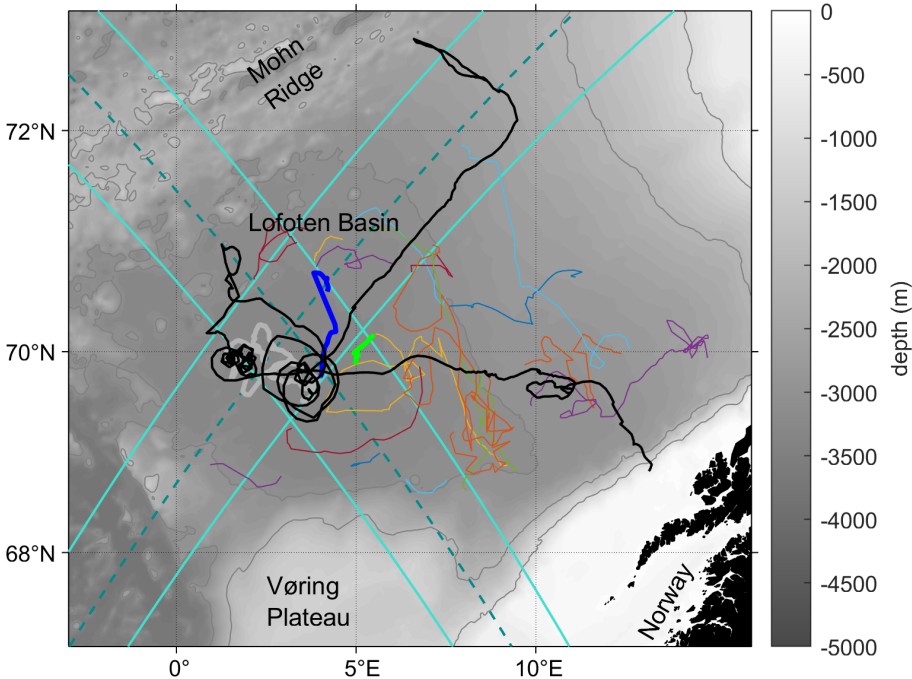

**Figure 2.** The Lofoten Basin, showing the path of SG563 (black), the position of the LV for the time period 1 January - 31 June 2023 (grey) taken from the altimetric Mesoscale Eddy Trajectories Atlas (META3.2exp NRT), and the outer edges of the Surface Water and Ocean Topography (SWOT) swathes (turquoise) and the SWOT nadirs (dark cyan). Bathymetry contours are every 1000 m. Coloured lines are the tracks of mesoscale eddies, also from META3.2exp NRT. We include all tracks where at least part of the track lies within the region 68.5 - 71.5°N, 0 - 10°E, in the time period 1 January - 30 June 2023. Two eddy tracks are highlighted which will be discussed further in the text: the bold blue line is Eddy A and the bold green line is Eddy B.

piloted, buoyancy-driven vehicles which profile to a maximum depth of 1000 m in a sawtooth pattern (Eriksen et al., 2001; Testor et al., 2019). SG563 carried a CT sail measuring conductivity and temperature. Sampling occurred every 10 s (∼1 m vertical resolution at typical vertical speeds of ∼0.1 m s$^{-1}$) in the upper part of the water column to 300 m, and every 15
135  s (∼1.5 m vertical resolution) below that. SG563 was deployed on the east side of the Lofoten Basin and proceeded west, arriving in the vicinity of the LV on 3 March. It completed approximately 200 dives in the vicinity of the LV in the period between 3 March and 7 May, then proceeded north-east before being recovered (Fig. 2).

The raw data were processed using the University of East Anglia Seaglider toolbox (https://bitbucket.org/bastienqueste/uea-seaglider-toolbox/). The Seaglider hydrodynamic flight model was tuned following Frajka-Williams et al. (2011). Depth-
average currents (DAC) were calculated from the difference between the glider's flight path found from GPS positions at the beginning and end of each dive, and the glider's flight path relative to the water as calculated using dead reckoning and the Seaglider hydrodynamic flight model. The thermal lag of the CT sail was corrected following the methods of Garau et al. (2011). Occasional poor quality data (e.g., from poor flushing of the conductivity cell when the glider is moving slowly) were





flagged and discarded, and additionally outliers in the salinity profiles were removed using a 3-run Hampel filter (Liu et al.,
2004); together these account for 3% of the total data collected. Conservative Temperature and Absolute Salinity were calculated using the thermodynamic equation of sea water (IOC et al., 2010), and all references to temperature and salinity from this point onwards should be taken to mean Conservative Temperature and Absolute Salinity. Potential density anomaly ($\sigma_0$) was calculated relative to the surface pressure.

Temperature and salinity were calibrated against available ship CTD data managed by the Institute of Marine Research in Bergen, Norway. CTD casts were chosen which were collected within 5 km and 3 days of a glider profile, the top 300 m was excluded to avoid the rapidly-changing effects of surface forcing, and data within any pycnoclines were also excluded. These selection criteria were chosen to maximise the number of profiles used for calibration while still ensuring that the matching profiles were sampling the same water masses. In particular, the selection criteria ensured that glider profiles within the LV were matched to CTD profiles within the LV, whereas glider profiles outside the LV were matched to CTD profiles outside the LV, since the water column properties inside and outside the LV are significantly different. This led to a total of 13 glider-CTD profile pairs used for calibration. Salinity was corrected by an offset of 0.021 g kg$^{-1}$, and temperature by an offset of 0.061°C.

## 2.2 Characterisation of the LV

Seagliders have a navigation mode which is well suited for piloting inside eddies: steering relative to the DAC of the previous dive. By steering at 90° to the previous dive's DAC, the glider will spiral in to the core of the LV which can be easily identified by pilots by the deep core of warm Atlantic Water. Once the core is reached, the glider can be commanded to steer at 270° to the previous dive's DAC, and will spiral out of the LV. These spiralling tracks, henceforth referred to as vortex realisations, take approximately 4 to 6 days to complete. This method has been used to study the LV previously (Yu et al., 2017; Bosse et al., 2019).

Each LV realisation aims to characterise the LV's hydrography and dynamics in radial section. First the glider's DACs are used to detect the LV centre by minimising a cost function applied to a rolling window of four consecutive DACs to find the position where DACs are most perpendicular to the vectors joining their positions to the eddy centre (Bosse et al., 2015). Detected centres are only kept when the glider is reasonably close to the LV's core where the cost function minimum is properly defined by closed contours. Geographical coordinates of glider sampling points are then transformed into cylindrical coordinates $(r, \theta, z)$ neglecting the eddy's ellipticity, with $r$ the radial distance from the LV centre, $\theta$ the azimuthal angle and $z$ the depth. As in Bosse et al. (2019), temperature, salinity and DAC data are bin-averaged on a regular grid (3 km in radial distance, 5 m in vertical) and optimally interpolated using correlation scales typical of the LV's radius ($L_r = 15$ km) and of the seasonal thermocline thickness ($L_z = 15$ m). The cyclogeostrophic balance is finally solved for azimuthal velocities (see appendix A in Bosse et al. (2016)), as the strong vorticity of the LV implies important effects of the non-linear centrifugal force.

The following LV characteristics are described for each realisation. The radius $R_v$ is defined as the radial distance of the velocity maximum $v_m$. Water properties of the inner core and outside the LV are defined as the mean profiles located at distance inferior to $2/3R_v$ and between $3R_v$ and $5R_v$, respectively. The LV's Eddy Kinetic Energy (EKE) is computed from





the volume integral from the centre to $1.5R_v$ and 0 to 1000 m of the kinetic energy density function: $0.5\rho_0 v_\theta^2(r,z)$ with $v_\theta(r,z)$ the azimuthal velocities and $\rho_0 = 1028$ kg m$^{-3}$ a reference seawater density, as in Bosse et al. (2019). LV vorticity is obtained in cylindrical coordinates using

$$\zeta(r,z) = \frac{1}{r}\left(\frac{\partial(rv_\theta)}{\partial r}\right), \tag{1}$$

and Rossby number $\mathrm{Ro} = \zeta/f$ as the vorticity, normalised by $f$, the Coriolis parameter. Potential Vorticity (PV, m$^{-1}$ s$^{-1}$) is finally calculated as

$$\mathrm{PV}(r,z) = \frac{1}{g}\left[N^2(f+\zeta) - \left(\frac{\partial v_\theta}{\partial z}\right)\left(\frac{\partial b}{\partial r}\right)\right], \tag{2}$$

with $b = -\sigma_0/(\rho_0 g)$ the buoyancy, $\sigma_0$ the potential density anomaly, $g = 9.81$ m s$^{-2}$ the gravitational acceleration, and $N^2 = \partial b/\partial z$ the buoyancy frequency.

## 2.3 Altimetry

Level-3 SWOT satellite altimetry data were retrieved in their last validated version (v1.02, CNES) during the Cal/Val phase from 29 March 2023 to 10 July 2023 for tracks 5 (descending pass) and 14 (ascending pass) from the AVISO ftp server (Dibarboure et al., 2024), as these two tracks have a cross-over in the LV region. Fields of noiseless Sea Surface Height Anomaly (SSHA) and Mean Dynamic Topography (MDT) at 2-km resolution were used to infer Absolute Dynamic Topography (ADT) following: ADT = SSHA + MDT. Here, we remove the monthly average of the ADT over the study area 68.5-71.5° N, 0-6° E, and refer to what remains as the local ADT anomaly (ADTa). Horizontal velocities $(u,v)$ satisfying the cylogeostrophic balance were computed using Jaxparrow open-source software (Bertrand et al., 2024). The variational approach of this new method allows to solve the gradient wind balance, where the iterative method previously developed by Penven et al. (2014) for larger scale gridded altimetry does not consistently converge when applied to high-resolution SWOT data. Relative vorticity maps were finally obtained by calculating

$$\zeta = \frac{\partial v}{\partial x} - \frac{\partial u}{\partial y}. \tag{3}$$

In order to refine the characteristics of the LV and the merging eddy during this critical period, we applied an eddy detection method to the SWOT-based high-resolution ADT. For each individual SWOT swath sampled during the Cal/Val phase, the maximum of ADT in the study area was considered as the LV centre position, $(x_{LV}, y_{LV})$. The parameters $(A^0_{LV}, A_{LV}, R_{LV})$ of an idealised Gaussian eddy

$$\mathrm{ADT}_{LV}(x,y) = A^0_{LV} + A_{LV}\exp^{-[(x-x_{LV})^2+(y-y_{LV})^2]/2R^2_{LV}} \tag{4}$$

were then derived from SWOT ADT. $A^0_{LV}$ was set as the median of ADT in the far-field considered as the region between 100 to 150 km away from the LV centre. $A_{LV}$ was set as the maximum of ADT (i.e., at the centre of the LV). Finally, R was inferred from a least-square regression between the observed ADT and the Gaussian model in a region of less than 45 km from



the LV centre. Note that this definition is consistent with a radius of maximum azimuthal velocity. This procedure was repeated on $\mathrm{ADT} - \mathrm{ADT}_{LV}$ in order to characterise the approaching eddy $[(x_B, y_B), A_B^0, A_B, R_B]$. (As will be discussed below, the approaching eddy is the eddy B whose track is shown in Fig. 2.) To avoid outliers and detection at the swaths' edge, these

estimates were not considered in the analysis when less than 75% of a disk of radius $R_{LV}$ or $R_B$ was covered by SWOT data, and the root mean square error between the Gaussian model and measured ADT was larger than 5 cm in a disk of 100 km radius. The eddies' dynamical intensity was then characterised in terms of maximum Rossby number at the eddy centre by resolving the cyclogeostrophic balance for the Gaussian model : $\mathrm{Ro}^{\max} = -1 + \sqrt{1 - 4gA_i/f^2R_i^2}$ for $\mathrm{Ro}_{\mathrm{geo}}^{\max} = -2gA_i/f^2R_i^2 > -0.5$ and $\mathrm{Ro}^{\max} = -1$ otherwise; with $i = (LV, B)$ and $\mathrm{Ro}_{\mathrm{geo}}^{\max}$ the maximum Rossby number in geostrophic balance. It is worth

noticing that $R_{LV}$ found with this method is about 30-35 km, which is larger than the glider-inferred $R_v$, but is coherent with previously reported LV radii inferred from satellite data. This larger radius from the satellite data than the in-situ glider observations is most likely due to the Gaussian fit smoothing the finer scale velocity signal of the core.

Eddy tracks were taken from the altimetric Mesoscale Eddy Trajectories Atlas (META3.2exp NRT, henceforth META3.2), which is produced by SSALTO/DUACS and distributed by AVISO+(https://www.aviso.altimetry.fr/) with support from CNES

(France), in collaboration with IMEDEA (Spain) (Mason et al., 2014; Pegliasco et al., 2022). This dataset uses data from previous generation altimetric satellites such as Jason, Sentinel and TOPEX/Poseidon and does not incorporate SWOT data. We show only those tracks where at least part of the track lies within the region 68.5 - 71.5°N, 0 - 10°E, in the time period 1 January - 30 June 2023 (Fig. 2). Two of these eddy tracks will be discussed in this paper: Eddy A, which was tracked from 17 February to 12 March and is shown in bold blue on Fig. 2, and Eddy B, which was tracked from 21 March to 4 April and is

shown in bold green on Fig. 2. Using this data allows us to extend the period in which we have data on eddy movements before the start of the SWOT Cal/Val phase.

## 2.4 Surface forcing data

Surface forcing data were taken from the ECMWF ERA5 reanalysis (Hersbach et al., 2023), and are shown in Fig. 3. ERA5 has a temporal resolution of one hour and a horizontal resolution of 31 km. Each of the surface forcing variables was averaged over

the area $69.5 - 70°$N, $2.5 - 4.1°$E (i.e., over 21 grid points), covering the typical variability in the location of the LV. This also agrees well with the vortex centre locations found in our data (see below). Mean net surface heat flux ($Q_{\mathrm{net}}$, in $\mathrm{W\,m^{-2}}$) was then calculated as the sum of the short-wave, long-wave, sensible and latent heat fluxes. The mean net surface freshwater flux ($\mathrm{FW}_{\mathrm{net}}$, in $\mathrm{kg\,m^{-2}}$) was calculated as the mean total precipitation rate minus the mean evaporation rate, and the magnitude of wind stress $|\tau_{\mathrm{w}}|$ ($\mathrm{N\,m^{-2}}$) was calculated as

$$|\tau_{\mathrm{w}}| = \sqrt{\tau_e^2 + \tau_n^2}, \tag{5}$$

where $\tau_e$ and $\tau_n$ are the mean eastward and mean northward turbulent surface stresses, respectively.



## 3 Results

SG563 visited the LV four times in the period March - May 2023, for a total of eight vortex realisations. Three of these realisations will be the focus of this paper: realisation 1, sampled between 5 and 14 March, which shows the late winter

properties of the LV, realisation 3, sampled between 12 and 15 April, which is the period when an eddy merged into the LV, and realisation 5, sampled between 18 and 22 April, which shows the effect of the merging. We do not show the other realisations because they do not add to the results presented here. In particular, because the glider spirals into the LV on each odd-numbered realisation and then out of the LV on the following even-numbered realisation, the glider dives in the centre of the LV at the start of each even-numbered realisation follow immediately after those at the end of the preceding odd-numbered realisation, and

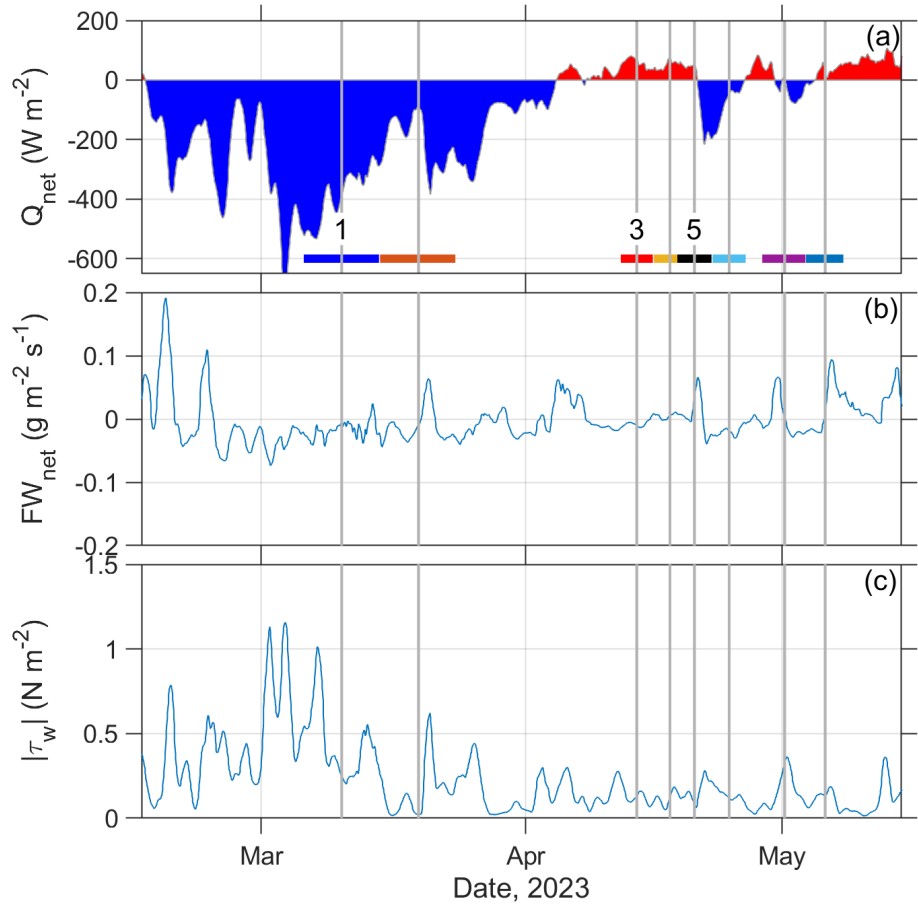

**Figure 3.** Surface forcing from ERA5, as described in section 2.4. Daily averaged (a) net surface heat flux, (b) net freshwater flux, and (c) wind stress. Vortex realisations are marked by the coloured bars along the bottom of panel (a), and realisations 1, 3 and 5 are labelled. On all panels, grey vertical lines mark the mean date of each vortex realisation.





the properties of the LV core do not change significantly in such a short space of time. Core profiles from each even-numbered realisation are thus extremely similar to the core profiles from the preceding odd-numbered realisation, and are not shown.

    In realisation 1, (Fig. 4, first column) the LV displays a well-mixed winter core with weakly stratified waters extending to 1000 m. Core profiles of temperature, salinity and PV are vertically uniform (Fig. 5), and the density is well-mixed throughout the water column to 1000 m (Fig. 6). The core of the LV (radial distance of about 10–15 km in Fig. 4), is surrounded by a rim

of increased azimuthal velocity, reaching a maximum of 0.4 m s$^{-1}$ (Fig. 4g). Low PV is seen throughout the core from the surface to 1000 m (Fig. 4p). The shape of the low PV area is lens-like, but extends further out of the core between depths of approximately 50 and 500 m. The relative vorticity of the core is strongly anticyclonic, reaching $-0.57f$ (Fig. 4). Note this is not a double core vertical structure, such as many previous observations have seen (e.g., Yu et al., 2017; Fer et al., 2018; Bosse et al., 2019), but instead shows the result of winter mixing to at least 1000 m, similar to the earlier observations of Søiland and

Rossby (2013) and Søiland et al. (2016).





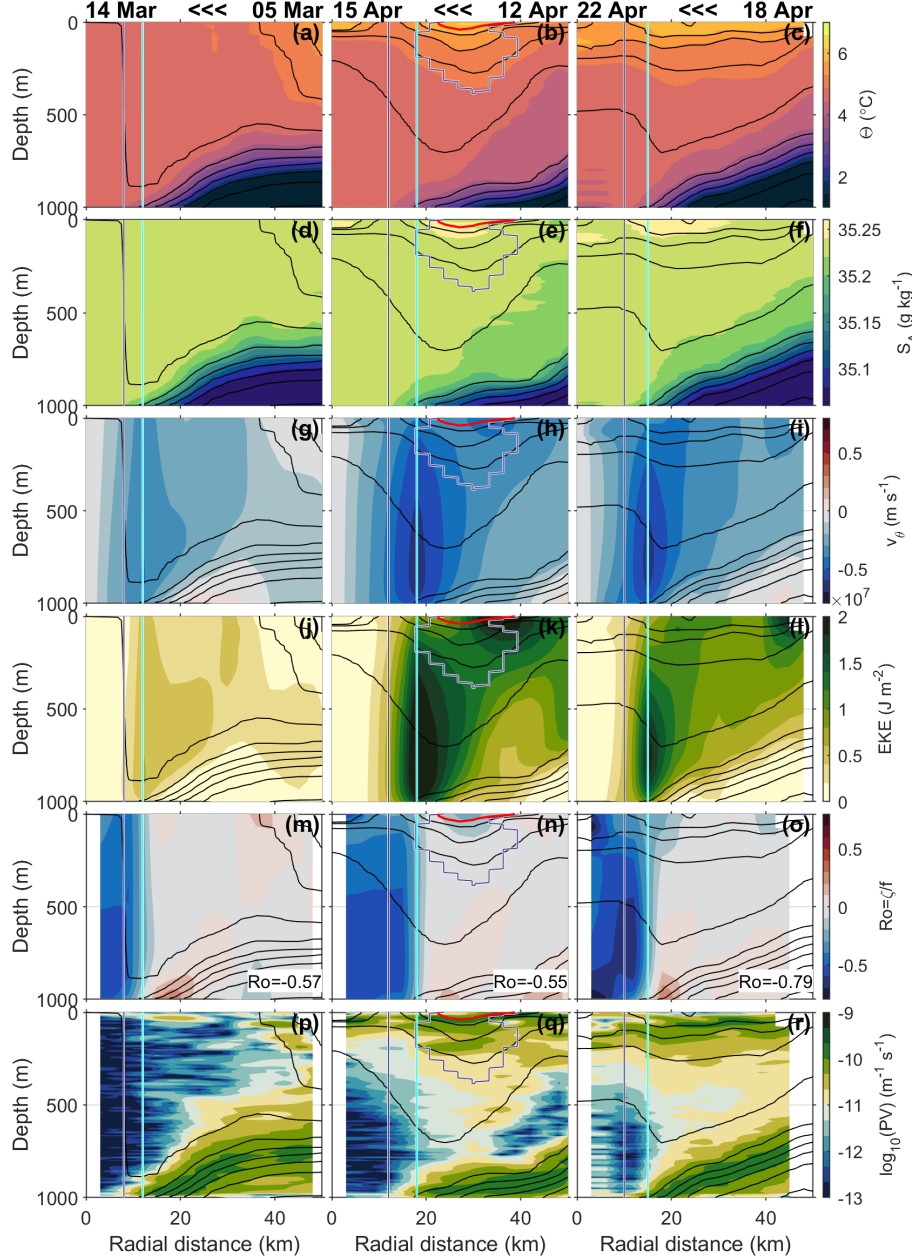

**Figure 4.** Sections through the LV: first column, realisation 1 between 5 and 14 March, second column, realisation 3 between 12 and 15 April, third column, realisation 5 between 18 and 22 April. Note that because the glider is travelling from outside the LV towards the centre of the LV in each of these realisations, the left hand side of each panel is later in date than the right hand side, as indicated by the labels at the top of each column. Top row, temperature; second row, salinity; third row, azimuthal velocity; fourth row, eddy kinetic energy; fifth row, Rossby number, $\mathrm{Ro} = \zeta/f$, with the minimum $\mathrm{Ro}$ value indicated bottom right; sixth row, potential vorticity, PV. Black contours on all panels show potential density anomaly ($\sigma_0$) every 0.04 from 27.6 to 28.0 kg m$^{-3}$. Vertical cyan lines on all panels show the radius of the core, $R_c$. Vertical blue lines show $2/3R_c$. In the second column, the blue contour outlines Eddy B, which is merging into the LV, and the red contour marks the $\sigma_0 = 27.6$ kg m$^{-3}$ isopycnal.





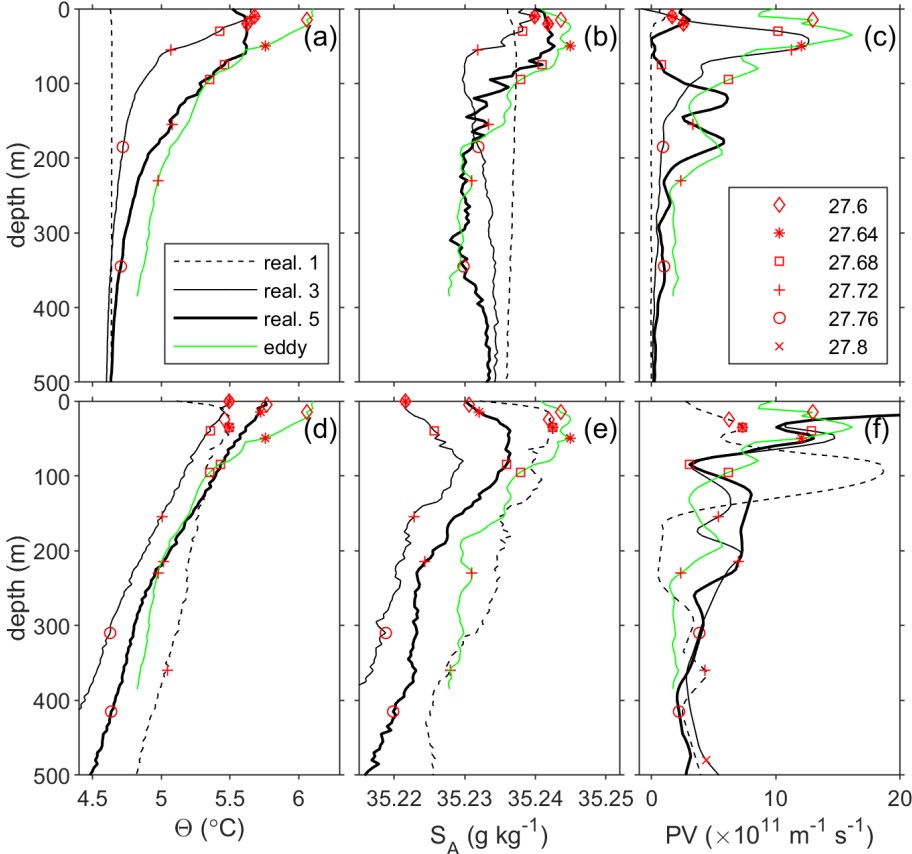

**Figure 5.** Average water profiles in the core of the LV (top row) and outside the LV (bottom row). First column, temperature, second column, salinity, third column, PV. (PV is multiplied by $10^{11}$ for display purposes.) In each panel, dashed black lines show properties during realisation 1, 5 - 14 March, thin black solid lines show properties during realisation 3, 11 - 15 April, and thick black lines show properties during realisation 5, 18 - 22 April. Green lines show the properties of the incoming eddy, as defined in the main text. Markers depict different potential density anomalies on each profile, as given by the legend on panel (c). There are no markers on the core profile for realisation 1 because the entire profile lies between $\sigma_0 = 27.772$ and $\sigma_0 = 27.773 \, \mathrm{kg \, m^{-3}}$.

Outside the LV, profiles are typical of the Lofoten Basin, with a thin (approximately 20 m), fresh and cold surface layer and somewhat warmer and saltier waters beneath, which then gradually cool and freshen towards 1000 m (Fig. 4). The change in properties is almost density-compensating from the surface to 100 m, as seen in figure 6, and even to 400 m the density changes are small.





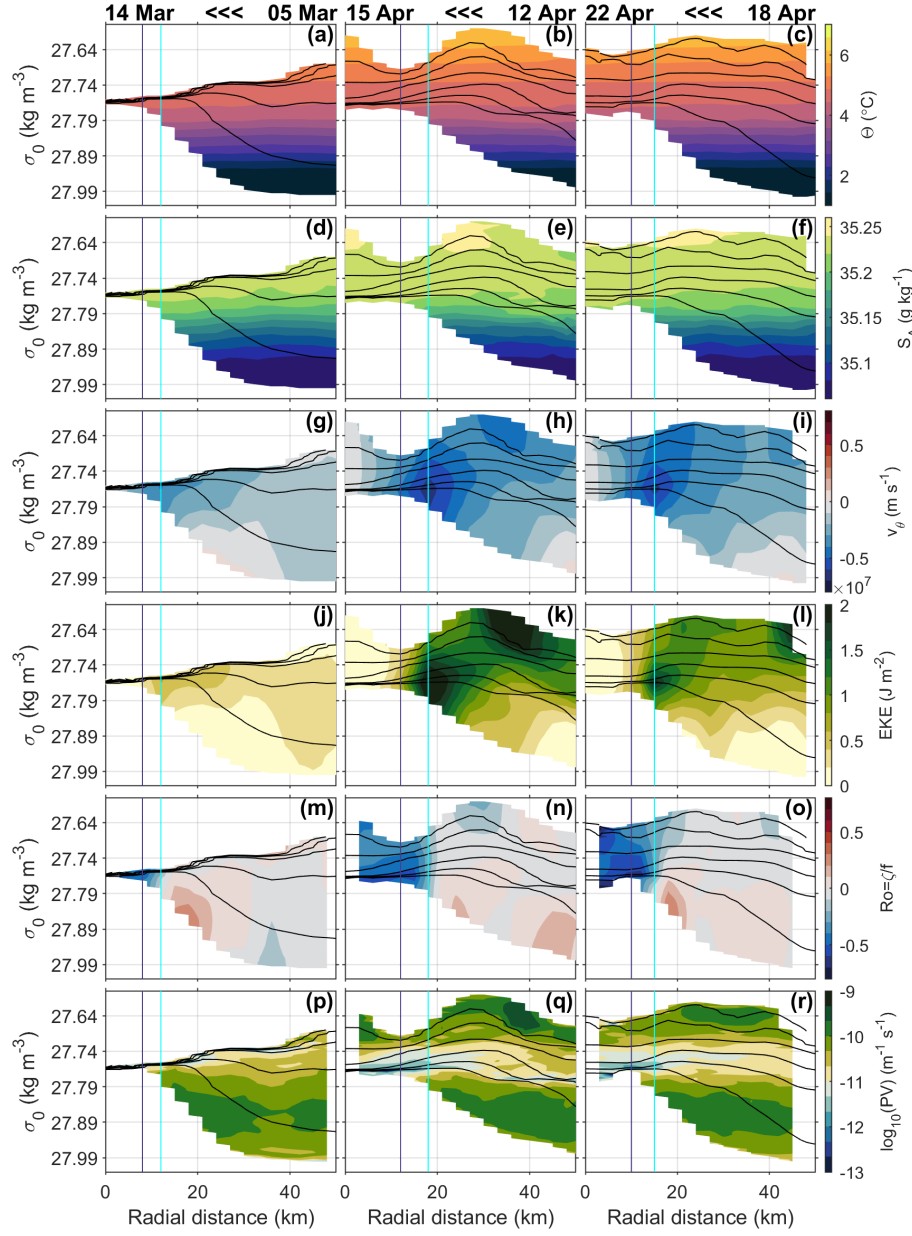

**Figure 6.** As figure 4, but against density instead of depth. Black contours on all panels are depth contours, at depths of 50, 100, 200, 400, 600 and 800 m, except in column 1, which does not show the 50 m contour because of the minimal density change between the surface and 100 m. Note that the y-axis scale on all panels is expanded by a factor of two between densities of 27.74 and 27.99 kg m$^{-3}$.



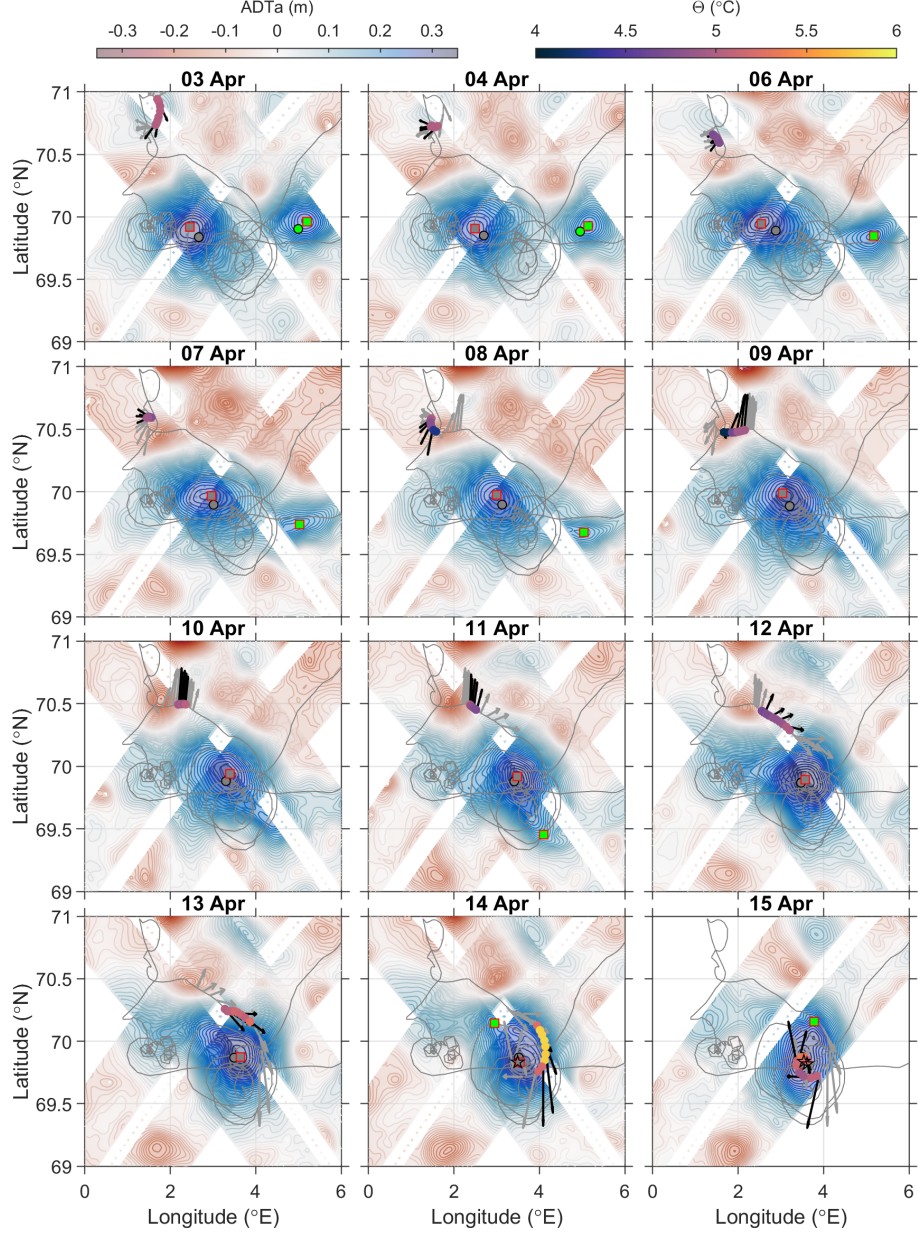

**Figure 7.** ADTa from the SWOT altimetry (coloured contours and shading) from 3 to 15 April 2023. Note that 5 April is not shown, but was extremely similar to 4 April. The grey line shows the whole-mission path of SG563, and coloured dots along this path show the average temperature of the top 100 m for profiles taken on the date given above each panel. Black arrows show the DACs measured that day, and grey arrows show the DACs for the day before and after. The black stars on 14 & 15 April show the LV position detected by the glider (section 2.2). Grey dots with a black outline mark the daily position of the LV from META3.2, and green dots with a black outline mark the daily position of Eddy B from META3.2, which is also shown as a green track on Fig. 2. The last date given for this track was 4 April 2023, thus its position is not shown on later dates. Grey squares with a red outline mark the position of the LV from the maximum ADT in the SWOT swath data, and green squares with a red outline mark the position of Eddy B from the SWOT swath data on days when the position met the eligibility criteria described in section 2.3.





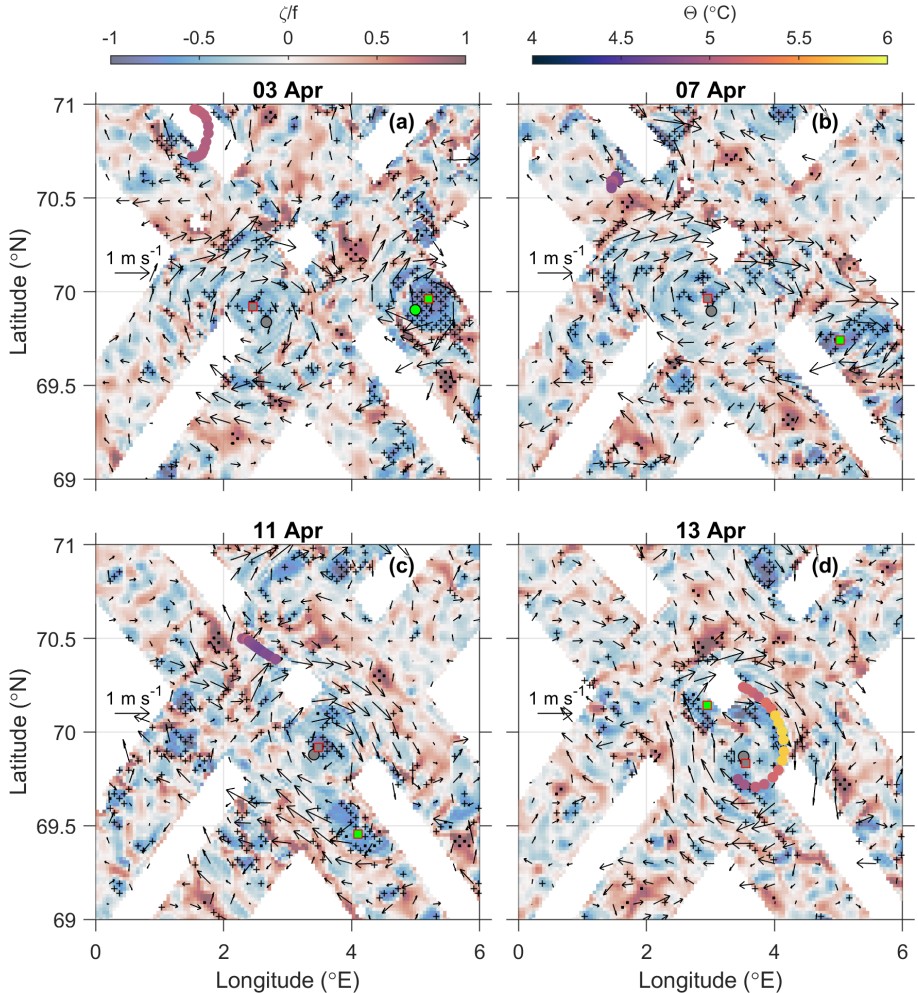

**Figure 8.** Relative vorticity calculated from the SWOT altimetry (coloured shading) on selected dates in April 2023. (Dates were selected simply to give an overview of the processes depicted, not because these dates are particularly significant.) This is the sum of the geostrophic and cyclostrophic components of vorticity. Black arrows show the surface velocity on those dates. Grey dots with black outlines, green dots with black outlines, grey squares with red outlines and green squares with red outlines are all as in Fig. 7. Coloured dots show the average temperature of the top 100 m measured by the glider, for a 48-hour period centred at noon of the date given above each panel. The vorticity is mostly dominated by the geostrophic component, but stippling marks where the cyclostrophic component is significant relative to the geostrophic component. Specifically, '+' stippling marks where the ratio of the cyclostrophic component to the total is $> 0.25$, and '.' stippling marks where the ratio of the cyclostrophic component to the total is $< -0.25$. Stippling is only shown where $|\zeta| > 0.4f$.

The LV is clearly visible as a persistent, approximately circular area of elevated ADTa (Fig. 7) and strongly anticyclonic vorticity (Fig. 8), centred between $69.75 - 70°$N, $2.5 - 4°$E (consistent with the LV positions in the literature), throughout the SWOT Cal/Val phase. The positions of the LV from META3.2 (grey dots with black outlines on Fig. 7) are close to the positions



of maximum ADT in the SWOT data (grey squares with red outlines on Fig. 7, henceforth SWOT-derived centre positions), with an average separation of 6.0 km in April 2023. On seven days (14-17 and 21-23 April 2023), positions for the centre

of the LV were also found from the glider data, as described in section 2.2. On these days, the average distance between the LV centre positions from META3.2 and the glider-derived LV centre positions was 4.8 km, and the average distance between the SWOT-derived and glider-derived LV centre positions was 4.9 km. At first glance, this might suggest that META3.2 gives comparable estimates of the centre position of the LV to the SWOT-derived centre positions. However, it is worth noting that, firstly, the sample size is small; secondly, the META3.2 positions are daily estimates, the SWOT-derived positions come from

the time of a particular pass of the SWOT altimeter, and the glider-derived positions are based on data from four dives covering a period of approximately 20 hours, and thus they do not all represent equivalent time periods; thirdly, the glider-estimated centre positions rely on an assumption of axial symmetry, but there are some slight distortions from circular in the ADTa contours (Fig. 7) which will cause some slight alterations in the glider-estimated centre positions. Given these provisos, it seems reasonable to conclude that the META3.2, SWOT-derived and glider-estimated centre positions are all within a few km

of each other, but we cannot say that one position estimate is better than the others during the limited period considered.

At the beginning of the SWOT Cal/Val phase, a smaller anticyclonic eddy is also visible to the east of the LV (Figs. 7 and 9). This eddy will be termed Eddy B from now on. Eddy B has been gradually approaching the LV over the period from 21 March to 4 April, as shown by its track from the META3.2 data (thick green track in Fig. 2). Eddy B is a region of very high anticyclonic vorticity where the cyclostrophic component is significant compared to other areas (Fig. 8). Eddy B gradually

elongates (7-8 April), and becomes attached to the southeastern side of the LV (9 - 12 April), following an approximately circular path around the LV centre (Figs. 7 and 9). Indeed, the evolution of the centre positions of Eddy B and the LV (Fig. 9) suggest that it is not simply that Eddy B travels around the LV, but that the LV and Eddy B corotate, as was previously found in idealized numerical studies (Yasuda and Flierl, 1997; Von Hardenberg et al., 2000; Carton et al., 2016). This elongation, attachment and co-rotation is similar to the merging process described by Trodahl et al. (2020). There are no valid detections

of Eddy B in the SWOT data between 11 and 14 April, but it seems to have accelerated considerably and wrapped around LV between those dates. Drifting along a circle at about 50 km from LV centre in 3 days, Eddy B would have travelled at speed comparable to the LV swirling velocities of about 0.6 m s$^{-1}$ during this period. It is possible that some of Eddy B is ejected to the west around 12 to 13 April, but we cannot be certain as that area is not covered by the glider observations. Eddy B begins to merge into the LV on 13 - 14 April as the distance between the two centre positions drops below about 50 km, but is still visible

as a distortion (on the northern side of the LV at a distance of approximately 30 km from the LV centre), in the otherwise fairly circular ADTa contours around the LV (Figs. 7 and 9), and as a region of strongly anticyclonic vorticity (Fig. 8). The glider was then approaching the LV for realisation 3 (12 - 15 April), and thus on 13 April it recorded temperatures in the uppermost 100 m which are slightly elevated compared to the waters surrounding the LV. On 14 April the glider passed through Eddy B and recorded noticeably warmer temperatures in the uppermost 100 m, up to 6.2°C near the surface and nearly 6°C for the 0-100

m average. Eddy B is also seen clearly as less dense waters at around 30 km from the centre of the LV (Fig. 6), and displays high eddy kinetic energy (EKE) and higher PV than the LV (Fig. 4k and q). Finally, on 15 April, the glider reached the core



of the LV, which was slightly warmer (5.6°C near the surface) than the waters surrounding the LV (4.9°C near the surface) but not as warm as Eddy B, which suggests that Eddy B has not yet merged into the core of the LV (Fig. 4b).

The Gaussian fits to the LV and Eddy B ADT, described in section 2.3 and illustrated in Fig. 9, show the evolution of various properties of both the LV and Eddy B between 2 to 25 April (Fig 10). Prior to the merger, the LV and Eddy B have similar amplitudes ($A_{LV} \simeq A_B \simeq 0.25$ m), but thanks to its smaller radius Eddy B is significantly more strongly anticyclonic ($Ro^{max} \simeq [-1, -0.6]$ versus $[-0.5, -0.35]$ for the LV considering cyclogeostrophy or geostrophy only). As the distance between the LV and Eddy B decreases, the LV's amplitude and radius increase while Eddy B's amplitude and radius decrease, suggesting that the interaction between the two vortices is dissipating Eddy B while reinforcing the LV. The rapid change in these properties should be treated with some caution, however: as Eddy B and the LV become more elongated due to their mutual interaction, a circular Gaussian fit becomes less representative. The absorption of Eddy B by the LV, and potential ejection of a low-PV parcel from the LV, could also explain the transitional increase in the LV radius ($R_{LV}$) observed between 15 and 20 April, indicating the final stage of the merger. Eddy B's amplitude slowly decreases from 2 April as the two approach each other, but then drops faster from 7 April on as the distance between the LV and Eddy B becomes smaller than about 80 km, testifying to a stronger interaction between them. The merging process seems to lead to the dissipation of Eddy B, despite an apparently more dynamical initial Rossby number (close to $-1$ for cyclogeostrophic balance and $-0.6$ considering geostrophy only, Fig. 10-c, as also seen in Fig. 8) at the surface. In terms of Rossby number, Eddy B quickly loses its strong signature as the interaction with the LV begins and even before the actual merger with the LV core. It is also worth noting that the LV's cyclogeostrophic Rossby number seems to increase in absolute value from about $-0.5$ to $-0.6$ between 3 to 23 April.

Although the glider passed through Eddy B on 14 April and reached the LV core on 15 April, Eddy B has not yet reached the LV core on 15 April: the glider was moving faster than the eddy was merging. Hence the core profiles during realisation 3 are actually profiles from just before Eddy B merged in. The LV core is already more stratified than during realisation 1 (Fig. 5), with warmer temperatures near the surface and a much wider range in density than in realisation 1 (Fig. 6). It has been previously reported that during winters, a weakened PV gradient in the upper part of the LV between the core and the waters outside the LV reduces the dynamical barrier and facilitates the intrusion of warm waters from the vortex rim (Bosse et al., 2019). While one could speculate that the core and outer temperatures of realisation 1 (Fig. 5a and d) might mix laterally to generate the core temperatures of realisation 3 (and similarly for PV, Fig. 5c and f), the salinity in the core in realisation 3 is fresher than both the core and outer salinities of realisation 1 and thus cannot be formed from a mixture of the two. This strongly suggests another water mass has contributed to the evolution of the core structure. We speculate that another eddy (Eddy A) has already merged into the LV in March for the following reasons:

- From 14 March (when the LV core was observed at the end of realisation 1) to 15 April (when the LV core was observed at the end of realisation 3), the average net surface heat flux over the mean position of the LV (Fig. 3) was -107 W m$^{-2}$. A negative surface heat flux cannot explain an increase in temperature in the LV core, so there must have been another source of warmer water.





**Figure 9.** Maps of (first column) SWOT ADTa and (second column) reconstructed fields from Gaussian fits of the LV and Eddy B on selected dates in April 2023. (a,b) 3 April, (c,d) 7 April, (e,f) 11 April, (g,h) 14 April. ADTa contours are shown every 5 cm. The cyan dots are LV centre positions (i.e., daily maximum ADT in the study area) from 2 April to 14 April 2023, with the cross corresponding to the centre position on the date of the panel, and the coloured cyan circle corresponding to the radius of the Gaussian fit $R_{LV}$ on that date. Similarly, the green dots are centre positions for Eddy B from 2 April to 14 April 2023, with the cross corresponding to the centre on the date of the panel, and the coloured green circle corresponding to the radius of the Gaussian fit $R_B$.



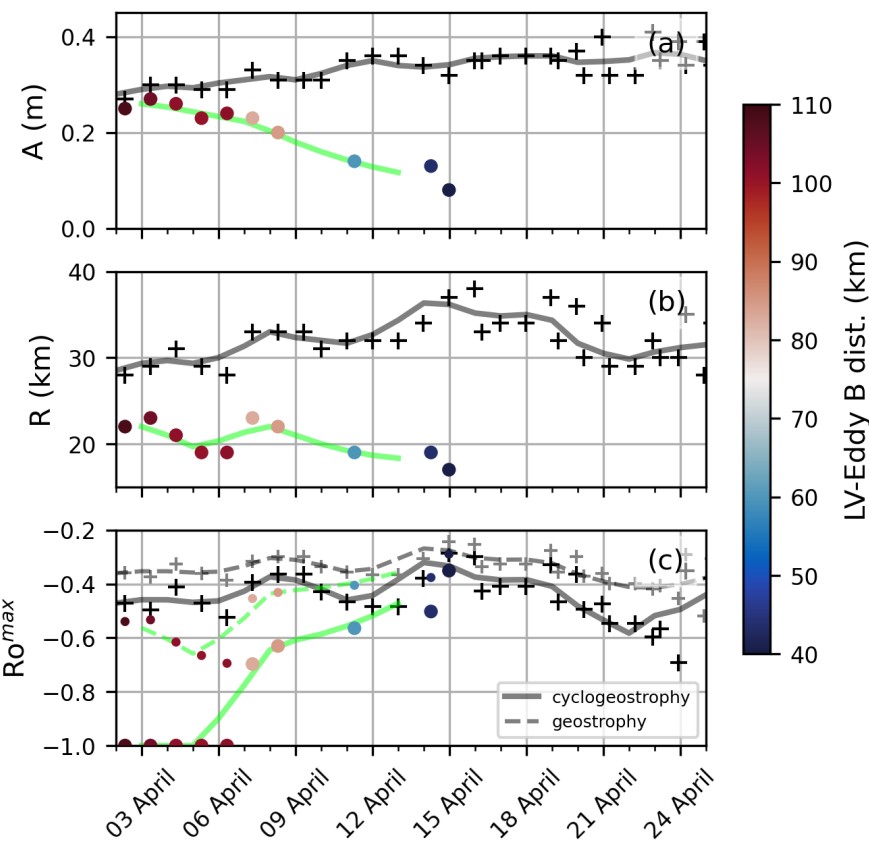

**Figure 10.** Temporal evolution of properties detected by SWOT during the April merger event between the LV and Eddy B: (a) Amplitude (b) radius from the Gaussian fitting and (c) maximum Rossby number $\mathrm{Ro}^{\mathrm{max}}$ at the LV or Eddy B centre. In all panels, dark grey crosses show values for the LV with a solid grey line showing the 3-day rolling average, and coloured dots show values for Eddy B with a solid green line showing the 3-day rolling average. The colour of the dots represents the distance between the centre positions of the LV and Eddy B. On panel (c), the paler grey dashed line and light grey crosses (for the LV), and dashed green line and smaller coloured dots (for eddy B) show geostrophic estimates, whereas the darker grey solid line, dark grey crosses, solid green line and larger coloured dots show cyclogeostrophic estimates (see section 2.3).



– Similarly, the average net surface freshwater flux from 14 March to 15 April was $-6.1 \times 10^{-6}$ kg m$^{-2}$ s$^{-1}$ (i.e., net evaporation). The core profile for realisation 3 is fresher than the profile for realisation 1 (Fig. 5), which cannot be explained by a negative freshwater flux, nor, as above, by intrusion of more saline waters from the vortex rim.

    – The META3.2 product shows another distinct eddy (Eddy A) approaching the LV earlier in the year (thick blue track on Fig. 2). This track runs from 17 February to 12 March, ending close to the average position of the LV, thus it may have

merged into the LV.

    – Finally, the LV appears to have ejected a parcel of low PV water, seen in Fig. 4q between approximately 400 - 800 m depth and 35 - 60 km from the centre of the LV. Note that this parcel of low PV water is almost invisible in Fig. 6 because its density range is extremely small. This agrees with the findings of Trodahl et al. (2020), who reported that some low-PV fluid was expelled from the LV during merging events in their modelling study. Moreover, Bosse et al.

(2019) used RAFOS float trajectories between 300 m and 800 m to document the Lagrangian coherence of the LV core - except during particular events where several floats were expelled at the same time, consistent with water parcels being expelled from the LV. However, the low PV parcel seen in realisation 3 was observed by the glider several days before Eddy B merged into the LV. Thus we speculate that this parcel of water may have been ejected during an earlier merger.

Since we lack both in situ observations and the high resolution SWOT altimetry during this period, we cannot be certain that

Eddy A merged into the LV between realisation 1 and realisation 3. It seems, however, the most plausible explanation for the changes in water properties in the LV core, and we will henceforth refer to this as the 'probable merger event in March'.

To assess the properties of Eddy B, observed in April, we use a simple definition to distinguish it from surrounding waters: water with temperatures which are at least 0.2°C warmer than the temperature of the LV core at the same depth are taken to be within Eddy B. This is shown as a blue contour on Fig. 4. The properties of Eddy B are calculated as an average in each

depth bin over the waters enclosed by this blue contour, and are shown in green in Fig. 5. By the time of realisation 5 (18 - 22 April), the LV core profiles to 400 m are similar to those observed in Eddy B a week earlier (Fig. 5). Although Trodahl et al. (2020) found that incoming eddies tended to be lighter than the LV and thus stacked on top of it during mergers to create a double-core vertical structure, we do not observe that here. This is because Eddy B is not significantly lighter than the LV: the probable merger event in March has already reduced the density of the upper layers of the LV. In the second column of Fig. 4, a

red contour marks the minimum potential density observed in the LV core during this realisation ($\sigma_0 = 27.6$), and is equivalent to the red contour in Trodahl et al.'s Fig. 14. The example of an incoming eddy shown in Trodahl et al.'s Fig. 14 is lighter than the LV to a depth of approximately 600 m but in realisation 3 Eddy B is only lighter than the lightest water in the LV core to a depth of 40 m. Eddy B therefore simply mixes with the LV rather than stacking on top of it, as also happens to eddies of the same density as the LV in the work of Trodahl et al. However, Eddy B's signature in temperature extends to approximately

400 m depth and thus it is lighter than the LV core at the same depth, as seen by the deepening of the isopycnals in Eddy B compared to the LV core (Fig. 4, second column). Water in the LV core is therefore pushed downward during the merger, as seen by the lowered isopycnals in the LV core in realisation 5 compared to realisation 3 (Fig. 4). We therefore see the vortex



intensification and change in relative vorticity predicted by Trodahl et al.: in realisation 5, after the merger, the LV's vorticity is even more strongly anticyclonic than in realisation 1, reaching $-0.79f$ (Fig. 4o).

As with the probable merger event in March, we consider whether surface forcing could have brought about the changes in the core properties observed in April. It has not been possible to determine this for salinity because the net change in core salinity was extremely small in the period from 15 to 22 April (i.e., between realisations 3 and 5); a net freshening of only 0.002 g kg$^{-1}$ over the full water column observed. The average net surface freshwater flux was $9.3 \times 10^{-6}$ kg m$^{-2}$ s$^{-1}$, also small, but in combination with the eddy merger, lowering of isopycnals, and possible leakage from the area outside the vortex, it was

not possible to separate the different effects. The situation is simpler for temperature because both Eddy B and the average net surface heat flux acted to warm the LV. However, the average net surface heat flux of 23 W m$^{-2}$ is an order of magnitude too small to bring about the observed temperature change in the LV core, hence the effect of Eddy B merging is dominant.

## 4    Discussion and conclusions

Conservative estimates of spin-down time scale for the LV (Fer et al., 2018; Bosse et al., 2019) require that such a long-

lived vortex must be maintained by sources that energise it. Merging of mesoscale eddies into the LV has been discussed in the literature as a potentially important mechanism; however, observations in support of merging events have been limited to surface signatures from satellite data (e.g., Raj et al., 2015). Our study, for the first time, presents a relatively coherent evolution of the surface and subsurface structure during a merging event by combining in situ glider observations with SWOT satellite altimetry data.

The observed merging process closely resembles the modelled process described by Trodahl et al. (2020). As observed at the surface by SWOT, Eddy B appeared to be more energetic and slightly smaller than the LV. As the incoming eddy approaches the LV and the two begin to interact, Eddy B gradually elongates and becomes attached to the side of the LV, and the two begin to corotate. This scenario also resembles quasi-geostrophic shallow water numerical simulations (Pavia and Cushman-Roisin, 1990; Yasuda and Flierl, 1997; Von Hardenberg et al., 2000; Carton et al., 2016). Our observations do not have the resolution

to show the filamentation described by Trodahl et al. (2020), although the elongation of Eddy B does suggest that something of that nature may be occurring. Due to cloud cover, no sea surface temperature, nor ocean colour image, could be used during the merger event to document finer scale filaments. We do not observe a double-core vertical structure after the merger of Eddy B and the LV, but this is not unexpected since the density range of Eddy B is similar to the LV's density range prior to the merger. The spin up of vorticity and eddy kinetic energy are also analogous to the merging process modelled by Trodahl et al. The

clear similarities between the modelled and observed process demonstrates that high resolution models offer complementary insight to better understand details of such eddy processes. Eddy-eddy interaction remains however a complex problem vastly dependent on the initial conditions of the eddies (e.g., PV signature and their vertical structure). Model evaluation crucially relies on observing capacities of such events.

During the observed period, merging eddies were the dominant process affecting the evolution of the LV. While lateral

mixing between the vortex core and the outer rim may also have been a factor, it is clear that, with regard to the heat and salt




budgets of the LV core, vertical 1D processes due to atmospheric forcing were greatly outweighed by the eddy merger events. In particular, the merger with Eddy A in March substantially altered the water mass properties of the LV, a process continued by the merger with Eddy B in April. The influx of warm waters, spin-up of eddy kinetic energy and increasingly anticyclonic vorticity (Fig. 4) will contribute to the longevity of the LV. However, these observations are from a fairly brief period. We observed one merging event clearly, and inferred another, in the time span of two months. This suggests that merging into the LV may be a more common occurrence than found by previous modelling studies, which found 3-4 merger events per year with no clear seasonal bias (Trodahl et al., 2020); 3-4 merger events per year but with slightly more mergers occurring during the period February–May and none in November–December (Köhl, 2007); and one to two mergers per year with a distribution skewed toward wintertime (Belonenko et al., 2017). It is possible that merger events in the real ocean are more common than modelled, but their average impact on the LV's properties may be smaller than modelled, as hinted at by the differing impacts of the two merger events discussed here. Longer-term (over several years) in-situ observations are still needed to study the evolution of the LV's properties, rate of merger events and possible seasonality in merger events, and to quantify the relative importance of wintertime convection versus merging (Ivanov and Korablev, 1995a, b; Köhl, 2007; de Marez et al., 2021), as well as the role of surface forcing and low-PV parcel ejection from the vortex core (Bosse et al., 2019).

The depth of wintertime convection in the LV can vary greatly, from a few hundred metres in the years observed by Yu et al. (2017) and Bosse et al. (2019), to the more than one thousand metres observed by Søiland and Rossby (2013) and Søiland et al. (2016), and seen in the present study. It is not known how representative the characteristics of the eddy merger reported here would be for differing starting conditions such as differing mixed layer depths, or an initial LV which already contains vertically stacked cores. Future observations incorporating biogeochemistry would also be of interest. Deep winter mixing in the LV will increase the vertical flux of nutrients and the depth of oxygen-rich waters, creating conditions which might enhance biological activity in spring. However, deep mixed layers below the euphotic depth inhibit phytoplankton growth: for example, Hansen et al. (2010) found that the onset of spring phytoplankton blooms was delayed by an average of two weeks in the deep mixed layers of anticyclonic eddies in the Norwegian Sea. Merging into the LV might enhance the phytoplankton growth in such an eddy through increased nutrient availability or suppress it through deepening the mixed layer depth. Moreover, if merger events occur more frequently than previously believed, the phytoplankton community in the LV will likely be affected by repeated injections of different water masses carrying different phytoplankton with them.

The new SWOT satellite during its Cal/Val phase provided a truly unique data set able to document the dynamics of a merger with the LV in the Norwegian Sea. Without the wide swaths and higher resolution (compared to conventional satellite altimetry) of the SWOT data, our analysis would have been impossible. However, in order to fully understand the complex eddy-eddy interactions at play, the surface signature is not sufficient and the deep PV signature and circulations are important parameters to consider (Verron and Valcke, 1994). Thus the in-situ glider data, which allow for prolonged observations at relatively high resolution, were also a vital component of this analysis.



*Data availability.* All data presented in this study are openly available. Glider data are available from the Norwegian Marine Data Centre, DOI:10.21335/NMDC-440347600. The SWOT_L3_LR_SSH product, derived from the L2 SWOT KaRIn Low rate ocean data products

(L2_LR_SSH) (NASA/JPL and CNES), is produced and made freely available by AVISO and DUACS teams as part of the DESMOS Science Team project (last accessed 18/10/2024). META3.2 eddy tracking data is available from https://www.aviso.altimetry.fr/en/data/products/value-added-products/global-mesoscale-eddy-trajectory-product/meta3-2-exp-nrt.html (last accessed 14/08/2024). The surface forcing data from ECMWF is available from the Climate Data Store, https://cds.climate.copernicus.eu/datasets (last accessed 16/10/2023).

*Author contributions.* IF conceived and planned the experiment, GMD led the glider mission. All authors developed the analysis which
GMD performed. GMD wrote the paper with critical feedback from IF and AB. AB performed the analysis and produced figures related to SWOT. All authors discussed the results and finalized the paper.

*Competing interests.* One of the co-authors is a member of the editorial board of Ocean Science.

*Acknowledgements.* The glider was operated by the Norwegian national facility for ocean gliders (NorGliders) at the Geophysical Institute, University of Bergen. We thank the NorGliders team, and the scientists and crew of deployment and recovery cruises.

This work was supported by the Office of Naval Research Global, award number N62909-22-1-2023. GMD received funding from the European Union's Horizon 2020 research and innovation programme under the Marie Skłodowska-Curie grant agreement number 101034309. AB acknowledges the support by Centre national d'Études Spatiales (CNES) with the TOSCA project (GLISS), and by Centre National de la Recherche Scientifique (CNRS/INSU) with the LEFE project (DISTURB-SWOT). This work is a contribution to the SWOT Adopt-A-Crossover Consortium.



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
