# Peer review of "Merging of a mesoscale eddy into the Lofoten Vortex in the Norwegian Sea captured by an ocean glider and SWOT observations"

_EGUsphere, 2025_

## Author Response (AR1)

Response to Reviews on "Merging of a mesoscale eddy into the Lofoten Vortex in the Norwegian Sea captured by an ocean glider and SWOT observations", by Damerell et al.

Throughout this response, comments from reviewers are given in *black italic*, our responses are in green, and quotes from the paper are in blue.

**Reviewer 1**

I have read "Merging of a mesoscale eddy into the Lofoten Vortex in the Norwegian Sea captured by an ocean glider and SWOT observations", by Damerell et al. The manuscript details the successful observation of an eddy merging event within the Lofoten Vortex (LV) in April 2023, using data from an ocean glider and the SWOT satellite. The merger resulted in increased vorticity and eddy kinetic energy. This study provides in-situ evidence supporting the hypothesis that anticyclonic eddies from the Norwegian Atlantic Current contribute to the LV's heat, salt content, and energy. The findings suggest that eddy mergers play a dominant role in the LV's evolution and persistence, surpassing the influence of atmospheric forcing and lateral mixing.

**Scientific Significance:**

The manuscript presents new in-situ and satellite data that provide valuable insights into the mechanisms sustaining the Lofoten Vortex (LV). The findings support previous hypotheses that eddy mergers play a crucial role in maintaining the LV by offering compelling new evidence of such an event. The conclusions are well-supported by the data.

Thank you for your very positive and helpful review. We have attempted to address the points you raised which we believe has significantly improved the paper. Please see our responses below.

**Scientific Quality:**

The study employs a valid scientific approach with appropriate methods. The manuscript references relevant literature and provides substantial background information. The processing of glider and SWOT data appears thorough, and dynamical considerations, including cyclogeostrophic balance, are properly accounted for. The analysis effectively supports the conclusion that an eddy merger occurred during the study period.

However, the manuscript lacks an error analysis, which would significantly strengthen the study. Addressing potential sources of error—such as those in the glider data, cyclogeostrophic velocities, potential vorticity, and radial gradients—would enhance the manuscript's rigor. Including an error analysis would also provide valuable guidance for future studies integrating glider data with SWOT observations.

Thank you for this suggestion – we agree that this adds to our previous analysis. There are quite a number of potential sources of error:

• Uncertainty in the glider's temperature and salinity measurements.

- Uncertainty in the glider dive-average currents.
- Uncertainty in the glider-estimated centre position. Since the position is estimated from the dive average currents, this can be covered by the point above.
- The assumption of radial symmetry, which one could consider as an impact on the radius of the Vortex.

We now use Monte-Carlo simulations to investigate these various sources. This is described in a new section in the methods, with further details and complete set of results given in Appendix B. The section in the body of the text is as below:

**2.5 Error analysis of in situ data**

To assess the sensitivity of the derived variables (e.g., azimuthal velocity maximum, relative vorticity, Rossby number) to various sources of uncertainty, we conducted a series of Monte Carlo simulations (Mooney, 1997), each isolating a specific source of error. For the measurement uncertainties of temperature and salinity, we generated 1000 sets (i.e., temperature and salinity) of perturbed data fields by adding normally distributed errors, with a prescribed standard deviation, to each measurement point. Each set of perturbed fields is then analysed by applying transformation into cylindrical coordinates, binning, optimal interpolation and calculations as required to obtain the derived variables, exactly as in section 2.2, giving 1000 estimates of each derived variable. The measurement uncertainty of the DACs is treated in the same way. Assessing the errors introduced by the assumption of axial symmetry involved applying scaling factors to the radial distances from the centre of the LV to profiles and DAC measurements, and again generating 1000 realisations. These scaling factors were inferred from the vortex ellipticity seen in the SWOT data. Further details can be found in Appendix A. We report the mean, standard deviation and 95% confidence intervals of the resulting distributions obtained from the 1000 estimates for each source of uncertainty. Since the distributions of the derived variables calculated in the Monte Carlo simulations were approximately normal, the 95% confidence interval for each derived variable was simply found as the range between the 2.5th and 97.5th percentile of those distributions.

The results from this error analysis are detailed in Appendix A. The errors from the CT sail measurement uncertainty are trivially small, but those from the DAC measurement uncertainty and from the assumption of axial symmetry are significant. When stating errors within the body of this paper, we give a conservative error estimate using the largest standard deviation in each case. For vortex realisation 3, the largest standard deviation for the maximum azimuthal velocity is derived from the DAC measurement uncertainty. In all other cases, the largest standard deviation is found from the assumption of axial symmetry, specifically from assuming the glider's transect was aligned with the major axis of the LV ellipse. Thus the errors stated below may be considered to be the "worst case", and likely an overestimate of the true uncertainty.

**Presentation Quality:**

The manuscript's presentation could be improved. While the introduction provides excellent background information, its structure is disorganized, with topics presented in a scattered manner. The lack of clear connections between key points weakens the manuscript's motivation. A more coherent and structured introduction would greatly improve readability and

strengthen the overall impact of the study. I strongly encourage the authors to refine this section to enhance clarity and cohesion.

We have revised the Introduction extensively to improve the clarity and flow. Detailed changes can be found in the tracked version of the revised manuscript.

The figures in general are ok, with good text size and readability, but I found some things hard to understand:

Fig. 3 – This takes up a lot of real estate for useful but not essential information.

We are reluctant to cut this figure because the atmospheric forcing is an important part of the environmental conditions, particularly in explaining that the well-mixed vortex core deeper than 1000 m in realization 1 was due to convection and strong wind forcing in March, and also ruling out the effect of net surface freshwater and heat fluxes when discussing the effect of merger. In all, our paper is of typical length and reducing real estate is not necessary.

Fig 4 – Can the authors please discuss what smoothing methods they use? Why not just show raw or binned data here? I think the discrete colorbars smear the data resolution and this muddle the results.

The data shown is as discussed in section 2.2, lines 171-173 in the original manuscript, now lines 183-185.

As in Bosse et al. (2019), temperature, salinity and DAC data are bin-averaged on a regular grid (3 km in radial distance, 5 m in vertical) and optimally interpolated using correlation scales typical of the LV's radius (Lr = 15 km) and of the seasonal thermocline thickness (Lz = 15 m).

No further smoothing is applied. However, we have now added an extra figure in the appendices, figure B1, which shows the raw data from the glider before the optimal interpolation is performed. We hope this addresses your concerns.

With regard to the discrete colorbars, we are not entirely certain what you mean by "smear the data resolution". In general, continuous colour scales allow for seeing greater variance in data while discrete color scales can make it easier to see patterns in the data. With regard to this figure, we consider the discrete colour scales version to be easier to interpret. What smears the data resolution is perhaps our choice of contour interval, which again we consider appropriate for presenting our data, especially when we present objectively mapped, relatively smooth fields.

Fig 5 – the green line is difficult to see. So are the red markers. One suggestion would be to change the color of the green line, and possibly make marker sizes larger, or make them the same shape but different colors.

We now indicate Eddy B with an orange-red colour instead of green, and have made the line thicker. In fact, Eddy B is now indicated with the same orange-red colour on all relevant figures, i.e., figure 2 and figures 4 to 9. On figure 5, the density markers now match the colour of the lines they are on and are somewhat larger. We are reluctant to make the markers the same

shape but different colours for different densities because that would be more difficult for people who are colour blind to interpret.

Fig 6. I didn't find the figure useful and was difficult to read. I found that references to this figure in the manuscript were easier to see in Fig. 5 – I suggest removing this or putting it in the supplementary section.

As authors, we found this figure very useful to interpret our observations. There is also information relating to density-compensation, when compared to Fig 4, which we briefly discuss. We are reluctant to remove this figure entirely as some readers may find it easier to interpret than figure 5. However, we agree that it is not critical to have it in the main manuscript and have moved this figure to the appendices. (Ocean Science reserves supplementary material for items that cannot reasonably be included in the main text or as appendices, such as datasets, videos, very large images, etc, thus it is more appropriate to place such a figure in an appendix rather than in supplementary material.)

Eq 3- Do you use this? Or so you use the cyclogeostrophic vorticity from swot? Can you clarify this please?

This equation is the correct way to compute both geostrophic and cyclogeostrophic relative vorticity. The only thing that changes is to use either geostrophic or cyclogeostrophic velocities. We have clarified in the text that we use cyclogeostrophic velocities.

Eq. 5 – I don't think this needs its own equation.

We agree and have removed this from the paper.

**Reviewer 2**

This paper presents in situ observations of a vortex merger in the Lofoten Basin, based on a combination of glider and altimetric data. The contribution of this study to scientific progress is substantial and twofold. First, it provides a 3D in situ observation of an asymmetric vortex merger in the ocean with unprecedented resolution. Second, it offers new insights into the mechanisms that sustain the quasi-permanence of the Lofoten Vortex, confirming that vortex mergers play a crucial role, as previously suggested.

I recommend this article for publication in Ocean Sciences, with only minor revisions to improve the clarity of the presentation.

Thank you for your very positive and helpful review. We have attempted to address your comments below.

**General comments:**

-The manuscript does an excellent job of reviewing and presenting current knowledge on Lofoten Vortex dynamics. However, it lacks even a brief introduction to the broader context of vortex mergers. Since the 1980s, oceanographers have extensively studied vortex mergers (e.g., Polvani et al. 1989, https://doi.org/10.1017/S0022112089002016). This study is highly significant for that research community, as it provides a rare 3D in situ observation of a phenomenon usually examined through idealized simulations.

I recommend adding a short paragraph in the introduction to introduce this field of study, as well as incorporating a discussion of the present observations in the context of vortex merger research. The authors could refer to the literature cited in Carton et al. 2016.

Thank you for this suggestion – this was indeed a noticable gap in the Introduction. We have now added the following paragraph in the Introduction (lines 111-119):

Vortex merging is an active field of research, though mostly through numerical simulation given the difficulty of in-situ observations of such transient processes. During merging, two likesigned eddies (i.e., both anticyclonic or both cyclonic) come into close contact with each other and then merge to form either one larger eddy or two asymmetrical eddies, depending on initial conditions (see, for example, Dritschel, 2002; Meunier et al., 2002; Reinaud and Dritschel, 2002; Bambrey et al., 2007; Özugurlu et al., 2008). While most studies of this process consider isolated vortex mergers, i.e., omitting the influence of environmental factors such as neighbouring eddies or large scale currents, de Marez et al. (2020) deduced that merging is influenced by the  $\beta$ -effect and surrounding eddies. Efforts to include more physical effects in studies of vortex mergers are often impaired by the lack of observations of these processes. The LV offers an ideal location to capture in-situ observations of such events. However, prior to the current study, we not aware of any direct in-situ observations of a merger event.

-The manuscript's clarity could be improved by using shorter, more concise sentences throughout the text. Also please check the synthax and the vocabular you use for the vortex merger; examples, L301 "more strongly anticyclonic" is poorly said, or L304 "dissipating [...] while reinforcing[...]", no it's an exchange of mass.

We have attempted to modify the text as you suggest. In particular, the Introduction has been extensively rewritten to improve flow and clarity.

-Could the authors consider adding subsesction in the result section? It would greatly improve the clarity of the different messages of the manuscript

We have added subheadings as follows:

- 3.1 Late winter conditions in the LV
- 3.2 Comparing centre positions
- 3.3 Merger with Eddy B
- 3.4 Probable merger event in March

-In my opinion, if Figs. 5,6,7,8 were placed in appendix, the manuscript would gain clarity. Then, putting in regards Figs 4 and 9 would be easier, and enough to describe the merger.

We have moved figure 6 to the appendices because it showed the same data as figure 4, just against density instead of depth, and we only refer to that figure twice. However, we disagree with moving the rest of the figures because we think they are important to the narrative and readers would just be flipping back and forth between the appendices and the main text. The vertical profiles of (new) Fig 5 are very important in identifying and discussing the effect of the merger on hydrography. We are particularly fond of (new) Fig 6 as this figure is central to the story and shows the progression of the vortex and Eddy B in relation to ADTa fields and the glider observations.

**Specific comments:**

L7 (and all the manuscript): in-situ/in situ -> \textit{in situ}. Also italized all latin expressions (e.g., i.e.,...)

We are following the Ocean Science author guidelines, which say that common latin expressions such as et al., e.g. and in situ should not be italized.

Fig. 1: it could be useful to add a schematic of LV and small eddies shed from the NwASC in Fig. 1

We have added the mean location of the LV to figure 1 and indicated the eddy shedding from the NwASC.

Authors could use the newly released v2.0 of SWOT data for the study

We have updated our SWOT data to v2.0 and reproduced all the related figures.

A T/S diagram for each realisation could be added in Fig. 4, where the water masses "inside" LV and Eddy B are identified.

Thank you for this useful suggestion. Figure 4 is already slightly too large for the page, so it was not possible to add T/S diagrams to that figure. We decided instead to add T/S diagrams in two ways. Firstly we added T/S diagrams of the average profiles (core, outer and Eddy B) to figure 5 because figure 5 shows the average profiles in the core, Eddy B, and outer region, which can now be easily compared with the T/S diagrams of the average profiles. Secondly, we have added a new figure in the appendices, figure B2, which shows T/S diagrams for the data shown in figure 4, i.e., for realisations 1, 3 and 5 using the full data instead of average profiles.

The authors are not clear (or maybe I missed it) in exlpaining why the azimuthal velocity of the LV is way larger in realisation 3 than realisation 1

Thank you for pointing this out. This is likely because of the probable merger of eddy A in March, but we forgot to say so before. This has been added (lines 379-381):

The LV's azimuthal velocity increased considerably between realisation 1 and realisation 3, from  $0.4(\pm0.05)$  m s-1 to  $0.6(\pm0.07)$  m s-1 (Fig. 4j and k). This kind of "spin-up" is an expected outcome of an eddy merger (Trodahl et al., 2020).